# The effects of prenatal dietary supplements on blood glucose and lipid metabolism in gestational diabetes mellitus patients: A systematic review and network meta-analysis protocol of randomized controlled trials

**Sumanta Saha**[1]*, **Sujata Saha**[2]

1 Department of Community Medicine, R. G. Kar Medical College, Kolkata, West Bengal, India,
2 Department of Mathematics, Mankar College, Mankar, West Bengal, India

\* sumanta.saha@uq.net.au

## Abstract

### Background

Several randomized controlled trials (RCT) investigated antenatal dietary supplements' effect on gestational diabetes mellitus patients' fasting plasma glucose levels, glycated hemoglobin levels, homeostasis model assessment of- insulin resistance and β-cell function, quantitative insulin sensitivity check index for glucose, high-, low-, and very-low-density lipoprotein cholesterol levels, total cholesterol levels, triglyceride levels, and triglyceride to high-density lipoprotein ratio. However, an efficacy comparison across various dietary supplements and their co-supplements are unavailable for these outcomes. Therefore, a systematic review protocol is proposed here to make a network meta-analysis (NMA)-based juxtaposition across the following dietary supplements- vitamins, Myo-inositol, choline, minerals, probiotics, prebiotics, synbiotics, and omega-3 fatty acids.

### Materials and methods

A database search will ensue in the PubMed, Embase, and Scopus databases for RCTs testing the above, irrespective of their geographical origin. Data on population characteristics, compared interventions, and outcomes of interest will get abstracted from the studies included in the proposed review. Each of the reviewed studies will get appraised using the revised Cochrane tool. For each outcome, the comparative efficacy across interventions will be estimated in weighted or standardized mean difference using the frequentist method NMA and presented with their 95% confidence interval using league tables. By constructing network maps and comparison-adjusted funnel plots, a visual assessment of the inter-interventional relation and publication bias in each NMA model will happen, respectively. The best-ranked intervention prediction for respective outcomes will transpire using the surface under the cumulative ranking curve values. The Stata statistical software (version 16) will be used for analysis, and statistical significance will be determined at p<0.05 and 95% confidence interval.

**Funding:** The authors received no specific funding for this work.

**Competing interests:** The authors have declared that no competing interests exist.

## Trial registration

**PROSPERO registration number:** CRD42020214378.

## Introduction

Gestational diabetes mellitus (GDM) is a medical complication of pregnancy. It's defined as glucose intolerance of any degree that develops or gets detected for the first time during pregnancy [1]. In 2017, nearly 21.3 million live births occurred to hyperglycemia-associated pregnancies, and in 86.4% of these, the hyperglycemia was GDM-associated [2]. Depending on the diagnostic criteria used, the prevalence of GDM among pregnant females can vary between 4–18% [3]. The complications of GDM can be both short (e.g., cesarean section, pre-eclampsia, polyhydramnios in the GDM mothers and hypoglycemia and jaundice in their neonates) [1] and long-term (e.g., type 2 diabetes in the GDM mothers and obesity, glucose intolerance, and metabolic syndrome in the children of GDM mothers) [4].

Optimum glycemic control is crucial for better outcomes in GDM patients and their neonates [5]. Hyperglycemia occurs in GDM pregnancies due to inadequate insulin secretion in the latter half of the pregnancy [6–8]. Like type 2 diabetes, peripheral insulin resistance and decreased insulin secretion play roles in the GDM pathophysiology; however, the exact reasons for insulin dysfunction in GDM remain poorly understood [9]. Common markers used to monitor glucose homeostasis in GDM patients include fasting plasma glucose (FPG), glycated hemoglobin (A1c), homeostasis model assessment (HOMA) indexes, and quantitative insulin sensitivity check index (QUICKI). The FPG levels in pregnancies with GDM are usually higher than those with no glucose intolerance [6]. The American College of Obstetricians and Gynecologists endorses the following blood glucose level during pregnancy–FPG <95 mg/dL and one and two-hour postprandial blood glucose below 130–140 and 120 mg/dL, respectively [1]. Concerning A1c, the American Diabetes Association recommends its use in pregnancy with other glycemic markers as it's less sensitive than oral glucose tolerance tests [1]. HOMA of insulin resistance (HOMA-IR), an independent predictor of GDM [10], increases in GDM gestations [11]. The HOMA-IR values in GDM gestations can be higher than that of non-GDM pregnancies [6, 7, 11–15]. Then, there are beta-cell function markers, the HOMA of β-cell function (HOMA-B), the values of which can be lower in GDM gestation than in pregnancies with no glucose intolerance [6, 16]. Similarly, the QUICKI values can be lower in pregnancies with GDM than those with no glucose intolerance [11].

The GDM induced dyslipidemia (consistent with insulin resistance) [17] is also critical concerning the long-term cardiovascular and diabetes risk of the affected mother [18, 19]. In contrast to normal gestation, the triacylglycerol and low density lipoprotein levels are higher and lower in GDM pregnancies, respectively [20]. However, high density lipoprotein (HDL) and total cholesterol levels don't vary much between normal and GDM pregnancies [17].

Given the importance of glucose and lipid-related metabolic markers in GDM, several clinical trials have investigated these in prenatal dietary supplements receiving GDM patients. Such trials showed that some of these markers improved on antenatal supplementation of vitamin D with the following co-supplements- probiotics [21], omega-3 fatty acid [22], omega-6 fatty acid [23], and a combination of calcium, zinc, and magnesium [24, 25]. Likewise, Myo-inositol supplementation prenatally decreased HOMA-IR, insulin, and FPG levels in GDM mothers [26]. Despite the abundance of these trials, there is a shortage of rigorous and comprehensive meta-analytic comparisons of the blood glucose and lipid metabolism among different

prenatal dietary supplements in GDM patients. Some meta-analyses have chiefly concentrated on perinatal outcomes only [27–33]. The pairwise meta-analysis (PMA) articles on metabolic markers in GDM patients have primarily juxtaposed dietary supplements (like vitamin D and probiotics) with placebo recipients or its non-recipients, making between-supplement comparisons sparse [34–36]. Concerning network meta-analysis (NMA), the NMA models of a review article [37] contrasting the effect of different dietary supplements in NMA patients were limited to certain glycemic markers only (FPG, insulin, and HOMA-IR) and were not inclusive of A1c, QUICKY, and HOMA-B. The integration method of its intervention arms supplementing vitamin D as a co-supplement in NMA models remains unclear [37]. About dietary supplements' role on the metabolic profile of GDM patients, best known to us, no review article distinguished their effects between individual supplements and their co-supplements.

Given these limitations, we propose this systematic review and NMA protocol to compare the effect of different dietary supplements (vitamins, Myo-inositol, choline, minerals, probiotics, prebiotics, synbiotics, and omega-3 fatty acids) and their co-supplements on blood glucose and lipid markers in GDM patients.

## Methods and analysis

The proposed review is registered with the PROSPERO (registration no CRD42020214378) [38]. This report adheres to Preferred Reporting Items for Systematic Review and Meta-Analysis Protocols (PRISMA-P) (2015) reporting system (S1 File) [39].

## Eligibility criteria

**Inclusion criteria**.

1. **Study design:** Parallel arm randomized controlled trials (RCT) of any duration will get included in the proposed review.

2. **Participant's characteristics:** The eligible study participants would include pregnant women diagnosed with GDM during their ongoing pregnancy irrespective of their age and previous GDM history. The diagnostic criteria used to diagnose GDM and the treatment given for GDM management will get accepted as per the trialists.

3. **Intervention arm/s:** The treatment arm/s may receive ≥1 of the following prenatal oral dietary supplements–vitamin A, B6, C, D, E, and K, Myo-inositol, choline, calcium, iodine, magnesium, zinc, and omega-3 fatty acids [40]. Iron and folic acid will not be assessed as dietary interventions as these often form a part of routine antenatal care. Additionally, the trials testing probiotics, prebiotics, and synbiotics will get included in the review. The dosages and regimen of the dietary supplements given to GDM patients will get accepted as per the trialists.

4. **Comparator arm:** The comparator arm participants should not be receiving any of the dietary interventions stated above and may receive a placebo.

5. **Primary outcomes:** As existing screening and management guidelines of GDM chiefly concentrates on glycemic markers, we included the following as our primary outcomes of interest- [41–43]

   a. FPG

   b. A1c

  c. HOMA-IR

  d. HOMA-B

  e. QUICKI

6. **Secondary outcomes:** Our secondary outcomes of interest include the following lipid-related markers as their role in screening or management of GDM are not yet established-

  a. HDL

  b. Low-density lipoprotein

  c. Very-low-density lipoprotein

  d. Total cholesterol

  e. Triglycerides

  f. Triglyceride to HDL ratio

Trials reporting about any of these markers will be eligible for recruitment in the review. If there are ≥2 publications based on the same trial population data, one reporting a higher number of outcomes will be included in the proposed review.

  **Exclusion criteria**.

1. Trials on pregnant females with pre-existing diabetes like type 2 diabetes will not get included in the proposed review.

2. Trials in which the GDM patients received the dietary supplements in non-oral forms like parenterally will get excluded.

## Literature search

We will search the PubMed, Embase, and Scopus databases unrestricted to any geographic boundary for articles published in any language between 1964 (the first known GDM diagnostic criteria got introduced this year by O' Sullivan and Mahan) [44] and to date.

  An additional search for papers will transpire in the bibliography of publications read in full text. Following are the prospective terms to be used in the PubMed search, based on key themes of the context (GDM, clinical trial, and dietary supplements)- "gestational diabetes" OR GDM OR pregnanc* OR gestation* OR hyperglycemia OR "insulin resistance" OR "glucose intolerance" AND micronutrient OR nutrient* OR nutrition OR "dietary supplement*" OR supplement* OR vitamin OR mineral OR myo-inositol OR choline OR calcium OR iodine OR magnesium OR zinc OR "omega-3" OR "omega 3" OR probiotic* OR bacteria OR prebiotic* OR symbiotic*. Possible list of MeSH terms to be included during the PubMed search are "Therapy, Nutrition" [MeSH] OR "Medical Nutrition Therapy" [MeSH] OR "Nutrition Therapy, Medical" [MeSH] OR "Therapy, Medical Nutrition" [MeSH] AND "Dietary Supplements" [MeSH] OR "Food Supplementations" [MeSH] OR "Supplements, Food" [MeSH] OR "Nutraceuticals" [MeSH] OR "Nutriceuticals" [MeSH] OR "Herbal Supplements" [MeSH] AND "Diabetes, Gestational" [MeSH] OR "Pregnancy-Induced Diabetes" [MeSH] OR "Gestational Diabetes" [MeSH] OR "Diabetes Mellitus, Gestational" [MeSH]. Relevant filters will be used to concentrate the search on RCTs.

  The appropriateness of respective search strings will get asserted when at least three pre-identified clinical trials meeting the above inclusion criteria of the proposed review are

identifiable among the retrieved citations sorted relevancy-wise (detailed elsewhere with example) [32]. Identical search methods and terms will be used to search the other databases.

## Study selection

We will then upload the database search-retrieved citations to the Rayyan systematic reviews software [45] for duplicate publication elimination and skimming of the title and abstract of the remaining articles. Then, we will retrieve seemingly eligible and dubious articles in full text and subsequently read them to determine their eligibility for the proposed review. The list of articles excluded after full-text reading will be retained.

## Data abstraction

In a pre-piloted data abstraction sheet (S2 File; using Google form) [46], the following details of the reviewed trials will get abstracted primarily-

a. **Study details:** The last name of the first author, year of publication, trial's id, nation/s where trial/s got conducted, obtainment of ethical clearance and participant consent, and funding information.

b. **Population characteristics:** The number and the average age of participants in respective treatment arms, gender distribution, gestational age at which they got recruited in the study, and previous history of GDM.

c. **Interventions compared:** Regarding the tested interventions, their constituents, dosage, and regimen will be gathered for all intervention arms.

d. **Outcomes of interest:** All glucose and lipid metabolism markers of interest measured at the end of intervention period will be collected.

   Data for analysis will get abstracted in a separate form (S3 File).

## Risk of bias assessment

The risk of bias assessment for the following domains will transpire using the Revised Cochrane risk-of-bias tool for randomized trials (RoB 2)–bias due to randomization process, deviation from intended interventions, missing outcome data, outcome measurement, and selective reporting [47]. Using the signaling questions, the risk of bias of these domains will get judged. The recording of the responses to these questions can be any of the following based on the review authors' judgment- yes, probably yes, probably no, no, and no information. Finally, based on the responses to the signaling questions, we will categorize each of the domains stated above into low or high risk of bias or domain with some concerns. The detailed methodology is available elsewhere [47].

## Review authors' role

Three authors will conduct this review. The review authors will independently complete the study selection, data abstraction, and critical appraisal of the reviewed trials and mitigate conflicts in an opinion by discussing. For unresolved disagreements, third-party help will be sought.

## Data synthesis

**NMA.** For respective outcomes, we will compare the efficacy across different dietary interventions using a frequentist method NMA model utilizing the endpoint means and their

standard deviations (SD). Due to the continuous nature of the outcome data, the ES estimation will happen in the weighted mean difference or standardized mean difference depending upon the uniformity or non-uniformity of the measuring units, respectively [48]. Data from respective supplements and their co-supplemented forms will get added to the NMA models discretely to allow a distinction between their effects.

**Criteria for choosing outcomes eligible for NMA.** An outcome will get included in the NMA model when it meets the following criteria [32]-

1. Low risk of heterogeneity: A NMA will transpire for adequately powered PMA depicting low heterogeneity risk. A PMA-based heterogeneity evaluation will ensue for respective outcomes when the PMA model includes data from $\geq$20 studies and/or the mean sample size is $\geq$80 to ensure an adequately powered (80%) assessment [32, 49]. Heterogeneity determined at $p<0.1$ using the $Chi^2$ statistics [50] will get quantified using $I^2$ values. At $I^2$ values of 25%, 50%, and 75%, the heterogeneity will be categorized as low, moderate, and high, respectively [51]. A random-effect or fixed-effect model PMA (inverse variance method) will be conducted depending on clinical and methodological diversity across the trials [47]. The endpoint means and their SDs of respective intervention arms will be combined for muti-intervention-arm trials using the following formulae [50]-

$$\text{Combined mean} = \frac{(n_1 m_1 + n_2 m_2)}{n_1 + n_2} \tag{1}$$

Combined SD

$$= \sqrt{\left(\left((n_1 - 1)\, sd_1^2\right) + \left((n_2 - 1)\, sd_2^2\right) + \left(\frac{n_1 n_2}{n_1 + n_2}\right)\left(m_1^2 + m_2^2 - 2m_1 m_2\right)\right)/\left((n_1 + n_2) - 1\right)} \tag{2}$$

in these equations $n_1$, $n_2$, $m_1$, $m_2$, $sd_1$ and $sd_2$ denote sample sizes of intervention arm 1 and 2 of a clinical trial, average values of arm 1 and 2, and SD of $m_1$ and $m_2$, respectively.

2. The NMA models must form a connected network.

3. A network with a degree of freedom for heterogeneity to enable a random-effect consistency model fitting will qualify.

4. A network with a degree of freedom for inconsistency to enable inconsistency model fitting will qualify.

## Transitivity and consistency

To ensure the trials included in respective NMA models vary in the compared interventions primarily [52], data from trials testing oral supplements will only get included in the NMA models, as bioavailability depends on routes of administration.

We will use the local and overall inconsistency tests for a statistical evaluation of transitivity and accept the network consistency assumption if both tests are non-indicative of inconsistency.

**Network map.** Network maps will be constructed to assess the relationship between interventions in the NMA models. Their nodes will represent the interventions tested in an NMA model. The width of a node will increase as more participants receive that intervention. The width of the edges, i.e., the connectors between the nodes, will denote the number of trials comparing the adjoining interventions and will thicken as more trials compare these. If excessive crossing of lines produces complex network maps, we will simplify these by swapping treatment pairs using an iterative method [53].

**Obtaining SD in special circumstances.** If endpoint means are reported with standard error (SE) or 95% confidence interval (CI) instead of SD, the latter will be calculated using the formulae 3 and 4, respectively [50].

$$SD = SE \, x \, \sqrt{n} \qquad\qquad (3)$$

$$SD = \sqrt{n} x \frac{(\text{upper limit} - \text{lower limit})}{3.92} \qquad\qquad (4)$$

where $n$ denotes sample size and SE denotes standard error; 3.92 (2x1.96) SE is used for 95% CI; 3.29 and 5.15 will be used instead of 3.92 if reported in 90 or 99% CI, respectively [50].

If respective treatment arms constitute of small sample sizes (<60 participants), the CI values of 3.92, 3.29, or 5.15 will get replaced by a slightly larger value derived from the specific t distribution [50].

**League tables and ranking probabilities.** The respective NMA model's effect sizes and their corresponding 95% confidence intervals will be reported in league tables. The diagonal cells of these tables will represent the interventions included in the model. Depending on the outcome type, whether a positive (e.g., HDL) or negative (e.g., FPG) statistically significant ES determines the favorable effect, the comparative efficacy between two interventions will get determined.

We will predict the best intervention for outcomes with statistically significant ES (as suggested from the league tables) using the surface under the cumulative ranking curve [54]. These values can range between 0–100% with higher values denoting a better-ranked interventions. Additionally, we will make cumulative ranking plots for visual contrast between the estimated and predicted ranking probabilities [55].

**Risk of bias across studies.** As the trials included in the prospective review will have a comparator arm not receiving the interventions of interest [56], comparison-adjusted funnel plots will be used to assess publication bias. An asymmetric plot will suggest variation between studies with large and small sample sizes [57, 58].

**Sensitivity analysis.** Metabolic derangement often requires pharmacotherapy initiation (e.g., insulin) in GDM patients. Henceforth, to disentangle any effect of pharmacotherapy from dietary supplements, such drug-treated GDM patients' trials will get excluded from NMA models during an iteration of the preliminary NMA. Besides, the NMA will get iterated after eliminating any trial with a high RoB component to see if its incorporation affected the main NMA findings.

## Analytic tools

The PMA and NMA analyses will incorporate the use of the 'meta' and 'network' package of Stata statistical software version 16.0 (StataCorp, College Station, Texas, USA), respectively. Statistical significance determination will materialize at a p-value of <0.05 and a 95% confidence interval.

## Reporting of the completed review

The PRISMA statement guideline for NMA will be used for reporting of the proposed review [59].

## Confidence in cumulative evidence

For respective outcomes, the statistically significant favorable effect of a dietary supplement will undergo quality appraisal using the GRADE approach (GRADE Working Group (2004))

[60], and evidence will be graded into one of the following quality categories- high, moderate, low, or very low.

## Strengths of the proposed review

1. The proposed review is likely to be rigorous as it will meta-analytically compare RCTs, the highest level of epidemiological evidence. However, its ultimate strength will depend on the quality of the trials.

2. The NMA will provide statistical estimates on relative efficacy between interventions not compared in any trial.

3. As the dietary supplements and their co-supplemented forms will get incorporated into the NMA models as discrete interventions, these will help distinguish their effect on the glycemic and lipid profile of GDM patients.

## Weaknesses of the proposed review

1. As the eligibility criteria of this study restrict the proposed review to recruit RCTs only, evidence from other trial designs (e.g., single-arm trials) will not get reviewed.

2. As iron and folic acids are not the interventions of interest in the proposed study due to their universal use in pregnancy, we will be unable to ascertain their effects on the outcomes.

## Supporting information

**S1 File. PRISMA checklist.** Preferred Reporting Items for Systematic review and Meta-Analysis Protocols (PRISMA-P) 2015 checklist.
(PDF)

**S2 File. Proposed data abstraction form.**
(PDF)

**S3 File. Data abstraction form for analysis.**
(PDF)

## Author Contributions

**Conceptualization:** Sumanta Saha.

**Methodology:** Sumanta Saha.

**Supervision:** Sumanta Saha.

**Validation:** Sumanta Saha.

**Writing – original draft:** Sumanta Saha.

**Writing – review & editing:** Sumanta Saha, Sujata Saha.

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
