## [Decision Letter · Decision Letter 0]

10 Mar 2022

PONE-D-22-03658Efficacy juxtaposition of dietary supplements in blood glucose and lipid metabolism in gestational diabetes mellitus patients: A systematic review and network meta-analysis protocol of randomized controlled trialsPLOS ONE

Dear Dr. Saha,

Thank you for submitting your manuscript to PLOS ONE. After careful consideration, we feel that it has merit but does not fully meet PLOS ONE’s publication criteria as it currently stands. Therefore, we invite you to submit a revised version of the manuscript that addresses the points raised during the review process.

We look forward to receiving your revised manuscript.

Kind regards,

Antonio Simone Laganà, M.D., Ph.D.

Academic Editor

PLOS ONE

Journal Requirements:

Additional Editor Comments:

The topic of the manuscript is interesting. Nevertheless, the reviewers raised several concerns: considering this point, I invite authors to perform the required major revisions.

Reviewers' comments:

Reviewer's Responses to Questions

**Comments to the Author**

1. Does the manuscript provide a valid rationale for the proposed study, with clearly identified and justified research questions?

Reviewer #1: Yes

Reviewer #2: Yes

2. Is the protocol technically sound and planned in a manner that will lead to a meaningful outcome and allow testing the stated hypotheses?

Reviewer #1: Yes

Reviewer #2: Yes

3. Is the methodology feasible and described in sufficient detail to allow the work to be replicable?

Reviewer #1: Yes

Reviewer #2: Yes

4. Have the authors described where all data underlying the findings will be made available when the study is complete?

Reviewer #1: Yes

Reviewer #2: Yes

5. Is the manuscript presented in an intelligible fashion and written in standard English?

Reviewer #1: Yes

Reviewer #2: Yes

6. Review Comments to the Author

You may also provide optional suggestions and comments to authors that they might find helpful in planning their study.

Reviewer #1: My concerns with the study are the following:

1. The problem of heterogeneity I believe should be extensively described and discussed in the discussion section.

2. The authors should add the funnel plot.

3. The strength and limitations of the study should be described.

4. The text needs reviewing by a professional editor as the English used, although good, still is not satisfactory for publication.

5. It does not appear that a protocol has been pre-registered for this review (eg in PROSPERO). This is a concern as it introduces potential bias to the review and does not align with Cochrane guidance. The authors should note that a protocol was not registered, and discuss as a limitation in the discussion.

6. I suggest the author provide the full electronic search strategy for at least one database, including any limits used, such that it could be repeated, as a supplementary file.

7. Although you make appropriate assessments of study quality, you are not using study quality to report the results. The study quality should be used to provide a feel for whether your results derive from low quality evidence, moderate quality etc. Cochrane suggest the use of the GRADE approach to capture this element.

8. Results were duplicated in the text and tables/figures. I suggest avoiding any duplication in order to improve the readability of the manuscript.

9. I suggest extensive revise of the discussion with more explanation on the results of the current study. In discussion, many factors that could have influenced the results. Please also discuss other factors.

Reviewer #2: The authors present a systematic review protocol with a network meta-analysis of randomized clinical trials in which the effect of the use of individual supplements and their co-supplemented variants on the glycemic and lipid profile of pregnant women with gestational diabetes will be evaluated. The proposal is interesting, the manuscript was well written, contemplating the PRISMA-P checklist and I have small considerations that I consider important for it to be published.

The title I found confusing and I suggest that the authors change it to one that makes the main objective clearer such as Effect of dietary supplements used during prenatal care on lipid and glycemic profiles of pregnant women with gestational diabetes: A systematic review and network meta-analysis protocol of randomized controlled trials.

The authors correctly assume that "Since the articles published in the English language will get incorporated in the proposed review, trial published in any other languages will remain uncovered." However, today, with the ease of online translators, it no longer makes sense to limit the language of studies to be included in a protocol or even in a systematic review, since articles in complex languages such as Chinese can be translated and their results incorporated. I suggest that authors reconsider and think about not restricting language in their search strategy.

Finally, I realized that there will be several outcomes considered to assess the effects of supplements on glycemic and lipid profiles, but I did not see in the manuscript which the authors define as the main and the secondary ones, with a rational explanation for the choice, as requested by PRISMA-P: "List and define all outcomes for which data will be sought, including prioritization of main and additional outcomes, with rationale."

7. PLOS authors have the option to publish the peer review history of their article (what does this mean?). If published, this will include your full peer review and any attached files.

Reviewer #1: No

Reviewer #2: **Yes: **Ricardo Ney Cobucci

---

## [Author Response · Author response to Decision Letter 0]

18 Mar 2022

Response to reviewers’ comment

Dear Reviewers,

We would like to thank you for reading our manuscript and sharing your valuable comments. Our responses to each of your comments can be found below-

Reviewer #1: 

Reviewer’s comment:

My concerns with the study are the following:

1. The problem of heterogeneity I believe should be extensively described and discussed in the discussion section.

Authors’ reply: 

Thank you for your comment. We have described about heterogeneity assessment in the ‘Methods and analysis’ section beneath the subheading ‘Criteria for choosing outcomes for NMA.’ We are quoting the relevant text from the manuscript here for your kind reference- ‘A NMA will transpire for adequately powered PMA depicting low heterogeneity risk. A PMA-based heterogeneity evaluation will ensue for respective outcomes when the PMA model includes data from ≥20 studies and/or the mean sample size is ≥80 to ensure an adequately powered (80%) assessment.[32,49] Heterogeneity determined at p<0.1 using the Chi2 statistics[50] will get quantified using I2 values. At I2 values of 25, 50, and 75%, the heterogeneity will be categorized as low, moderate, and high, respectively.[51] A random-effect or fixed-effect model PMA (inverse variance method) will be conducted depending on clinical and methodological diversity across the trials.[47]’ 

Moreover, this is a protocol article and not the actual review; therefore, heterogeneity assessment across the trials was beyond the scope. 

Reviewer’s comment: 

2. The authors should add the funnel plot.

Authors’ reply:

Thanks for your comment. As this is a protocol and not the original review, there wasn't any data analysis associated with this manuscript. Henceforth, adding a funnel plot was beyond the scope. However, we have stated about constructing the comparison-adjusted funnel plots in the manuscript using the following text- ‘As the trials included in the prospective review will have a comparator arm not receiving the interventions of interest, comparison-adjusted funnel plots will be used to assess publication bias. An asymmetric plot will suggest variation between studies with large and small sample sizes.[56,57]’ Upon the commencement of the review, when data from the trials will be available for analysis, we will include these plots in supplementary files.

Reviewer’s comment:

3. The strength and limitations of the study should be described.

Authors’ reply:

 Thank you for the comment. We added the strengths and limitations. The relevant texts from the revised manuscript are quoted here for your reference-

‘Strengths of the proposed review

1. The proposed review is likely to be rigorous as it will meta-analytically compare RCTs, the highest level of epidemiological evidence. However, its ultimate strength will depend on the quality of the trials.

2. The NMA will provide statistical estimates on relative efficacy between interventions that were not compared in any trial.

3. As the dietary supplements and their co-supplemented forms will get incorporated in the NMA models as discrete interventions, these will help distinguish their effect on the glycemic and lipid profile of GDM patients. 

Weaknesses of the proposed review

1. As the eligibility criteria of this study restrict the proposed review to recruit RCTs only, evidence from trials of other designs (e.g., single-arm trials) will not get reviewed.

2. As iron and folic acids are not the interventions of interest in the proposed study due to their universal use in pregnancy, we will be unable to ascertain their effects on the outcomes.’

Reviewer’s comment:

4. The text needs reviewing by a professional editor as the English used, although good, still is not satisfactory for publication.

Authors’ reply:

 Thank you for the comment. Unfortunately, we don’t have the fund to use professional English language editing services. However, we edited the manuscript meticulously to improve it. If there are still any sentences or paragraphs, which you think need further amendment, please highlight those so that we can edit them further.

Reviewer’s comment:

5. It does not appear that a protocol has been pre-registered for this review (eg in PROSPERO). This is a concern as it introduces potential bias to the review and does not align with Cochrane guidance. The authors should note that a protocol was not registered, and discuss as a limitation in the discussion.

Authors’ reply:

 Thank you for the comment. The protocol is already registered with the PROSPERO (registration no. CRD42020214378). Please find it mentioned beneath the abstract and in the ‘Methods and analysis’ section. 

Reviewer’s comment:

6. I suggest the author provide the full electronic search strategy for at least one database, including any limits used, such that it could be repeated, as a supplementary file.

Authors’ reply:

 Thank you for the comment. The draft search strategy to be used for searching the PubMed database along with limits to be used can be found under the subheading ‘Literature search’ in the ‘Methods and analysis’ section. We quote the relevant text here for your reference- ‘Following are the prospective terms to be used for the PubMed search, based on key themes of the context (GDM, clinical trial, and dietary supplements)- "gestational diabetes" OR GDM OR pregnanc* OR gestation* OR hyperglycemia OR “insulin resistance” OR “glucose intolerance” AND micronutrient OR nutrient* OR nutrition OR “dietary supplement*” OR supplement* OR vitamin OR mineral OR myo-inositol OR choline OR calcium OR iodine OR magnesium OR zinc OR “omega-3” OR “omega 3” OR probiotic* OR bacteria OR prebiotic* OR symbiotic*. Possible list of MeSH terms to be included during the PubMed search are “Therapy, Nutrition” [MeSH] OR “Medical Nutrition Therapy” [MeSH] OR “Nutrition Therapy, Medical” [MeSH] OR “Therapy, Medical Nutrition” [MeSH] AND “Dietary Supplements” [MeSH] OR “Food Supplementations” [MeSH] OR “Supplements, Food” [MeSH] OR “Nutraceuticals” [MeSH] OR “Nutriceuticals” [MeSH] OR “Herbal Supplements” [MeSH] AND “Diabetes, Gestational” [MeSH] OR “Pregnancy-Induced Diabetes” [MeSH] OR “Gestational Diabetes” [MeSH] OR “Diabetes Mellitus, Gestational” [MeSH]. Relevant filters will be used to concentrate the search on RCTs.’

Reviewer’s comment:

7. Although you make appropriate assessments of study quality, you are not using study quality to report the results. The study quality should be used to provide a feel for whether your results derive from low quality evidence, moderate quality etc. Cochrane suggest the use of the GRADE approach to capture this element.

Authors’ reply:

 Thank you for the comment. We have mentioned evidence quality assessment beneath the ‘Confidence in cumulative evidence’ subheading under the ‘Methods and analysis’ section. We quote here the sentences addressing it from the manuscript- ‘For respective outcomes, the statistically significant favorable effect of a dietary supplement will undergo quality appraisal using the GRADE approach (GRADE Working Group (2004)),[59] and evidence will be graded into one of the following quality categories- high, moderate, low, or very low.’

Reviewer’s comment:

8. Results were duplicated in the text and tables/figures. I suggest avoiding any duplication in order to improve the readability of the manuscript.

Authors’ reply:

 Thank you for the comment. We would like to clarify that our manuscript has no tables or figures. Besides, as this is a protocol article, it didn't involve data abstraction and analysis. Therefore, results don't apply to this paper. However, upon completing the proposed review, we will remain cautious about not duplicating the findings stated in the main text in the figures and results.

Reviewer’s comment:

9. I suggest extensive revise of the discussion with more explanation on the results of the current study. In discussion, many factors that could have influenced the results. Please also discuss other factors.

Authors’ reply:

Thank you for the comment. As stated in reply to your previous comment, this is a protocol article; therefore, it doesn't have any results. Furthermore, we followed the PRISMA-P checklist, which recommends two key components in a protocol- the introduction and methods section. Henceforth, a discussion of the results is not relevant to this manuscript. We will keep your suggestions and implement them when the manuscript drafting of the finalized review commences. 

Thank you.

Reviewer #2: 

Reviewer’s comment:

The authors present a systematic review protocol with a network meta-analysis of randomized clinical trials in which the effect of the use of individual supplements and their co-supplemented variants on the glycemic and lipid profile of pregnant women with gestational diabetes will be evaluated. The proposal is interesting, the manuscript was well written, contemplating the PRISMA-P checklist and I have small considerations that I consider important for it to be published. The title I found confusing and I suggest that the authors change it to one that makes the main objective clearer such as Effect of dietary supplements used during prenatal care on lipid and glycemic profiles of pregnant women with gestational diabetes: A systematic review and network meta-analysis protocol of randomized controlled trials.

Authors’ reply:

 Thank you for the comment. We updated the title as the following- ‘The effects of prenatal dietary supplements on blood glucose and lipid metabolism in gestational diabetes mellitus patients: A systematic review and network meta-analysis protocol of randomized controlled trials’

Reviewer’s comment:

The authors correctly assume that "Since the articles published in the English language will get incorporated in the proposed review, trial published in any other languages will remain uncovered." However, today, with the ease of online translators, it no longer makes sense to limit the language of studies to be included in a protocol or even in a systematic review, since articles in complex languages such as Chinese can be translated and their results incorporated. I suggest that authors reconsider and think about not restricting language in their search strategy.

Authors’ reply:

 Thank you for the comment. We appreciate your suggestion and have decided not to restrict our search to the English language articles. The text representing it in the revised manuscript is quoted here for your reference- ‘We will search the PubMed, Embase, and Scopus databases unrestricted to any geographic boundary for articles published in any language between 1964 (the first known GDM diagnostic criteria got introduced this year by O’ Sullivan and Mahan)[44] and to date.’

Reviewer’s comment:

Finally, I realized that there will be several outcomes considered to assess the effects of supplements on glycemic and lipid profiles, but I did not see in the manuscript which the authors define as the main and the secondary ones, with a rational explanation for the choice, as requested by PRISMA-P: "List and define all outcomes for which data will be sought, including prioritization of main and additional outcomes, with rationale."

Authors’ reply:

Thank you for the comment. Beneath the subheading ‘Inclusion criteria’ in the ‘Methods and analysis’ section, we clarified the primary and auxiliary outcomes. We quote here the text from the revised manuscript for your reference- 

1. ‘Primary outcomes: As existing screening and management guidelines of GDM chiefly concentrates on glycemic markers, we included the following as our primary outcomes of interest-[41–43]

a. FPG

b. A1c

c. HOMA-IR

d. HOMA-B

e. QUICKI

2. Secondary outcomes: Our secondary outcomes of interest include the following lipid-related markers as their role in screening or management of GDM are not yet established-

a. HDL

b. Low-density lipoprotein

c. Very-low-density lipoprotein

d. Total cholesterol

e. Triglycerides

f. Triglyceride to HDL ratio’

Thank you.

---

## [Decision Letter · Decision Letter 1]

18 Apr 2022

The effects of prenatal dietary supplements on blood glucose and lipid metabolism in gestational diabetes mellitus patients: A systematic review and network meta-analysis protocol of randomized controlled trials

PONE-D-22-03658R1

Dear Dr. Saha,

We’re pleased to inform you that your manuscript has been judged scientifically suitable for publication and will be formally accepted for publication once it meets all outstanding technical requirements.

Kind regards,

Antonio Simone Laganà, M.D., Ph.D.

Academic Editor

PLOS ONE

Additional Editor Comments (optional):

I carefully evaluated the revised version of this manuscript.

Authors have performed the required changes, improving significantly the quality of the paper.

Reviewers' comments:

Reviewer's Responses to Questions

**Comments to the Author**

1. Does the manuscript provide a valid rationale for the proposed study, with clearly identified and justified research questions?

Reviewer #2: Yes

2. Is the protocol technically sound and planned in a manner that will lead to a meaningful outcome and allow testing the stated hypotheses?

Reviewer #2: Yes

3. Is the methodology feasible and described in sufficient detail to allow the work to be replicable?

Reviewer #2: Yes

4. Have the authors described where all data underlying the findings will be made available when the study is complete?

Reviewer #2: Yes

5. Is the manuscript presented in an intelligible fashion and written in standard English?

Reviewer #2: Yes

6. Review Comments to the Author

You may also provide optional suggestions and comments to authors that they might find helpful in planning their study.

Reviewer #2: The authors accepted most of the suggestions for improving the manuscript and left it ready to be published. Congratulations.

7. PLOS authors have the option to publish the peer review history of their article (what does this mean?). If published, this will include your full peer review and any attached files.

Reviewer #2: **Yes: **Ricardo Ney Cobucci

---

## [Editor Report · Acceptance letter]

22 Apr 2022

PONE-D-22-03658R1 

The effects of prenatal dietary supplements on blood glucose and lipid metabolism in gestational diabetes mellitus patients: A systematic review and network meta-analysis protocol of randomized controlled trials 

Dear Dr. Saha:

I'm pleased to inform you that your manuscript has been deemed suitable for publication in PLOS ONE. Congratulations! Your manuscript is now with our production department. 

Kind regards, 

on behalf of

Dr. Antonio Simone Laganà 

Academic Editor

PLOS ONE